# Application of Mediterranean Diet in Cardiovascular Diseases and Type 2 Diabetes Mellitus: Motivations and Challenges

**DOI:** 10.3390/nu14132777

**Published:** 2022-07-05

**Authors:** Najwa Salim AlAufi, Yoke Mun Chan, Mostafa I. Waly, Yit Siew Chin, Barakatun-Nisak Mohd Yusof, Norliza Ahmad

**Affiliations:** 1Department of Dietetics, Faculty of Medicine and Health Sciences, Universiti Putra Malaysia, Seri Kembangan 43400, Selangor, Malaysia; gs51039@student.upm.edu.my (N.S.A.); chinys@upm.edu.my (Y.S.C.); bnisak@upm.edu.my (B.-N.M.Y.); 2Department of Food Science and Nutrition, College of Agricultural and Marine Sciences, Sultan Qaboos University, Al-khod 50123, Oman; mostafa@squ.edu.om; 3Department of Community Health, Faculty of Medicine and Health Sciences, Universiti Putra Malaysia, Seri Kembangan 43400, Selangor, Malaysia; lizaahmad@upm.edu.my

**Keywords:** mediterranean diet, cardiovascular disease, type 2 diabetes mellitus, motivation, challenges

## Abstract

Objective: Cardiovascular disease (CVD) is the leading cause of disability and death in many countries. Together with CVD, Type 2 diabetes mellitus (T2DM) accounts for more than 80% of all premature non-communicable disease deaths. The protective effect of the Mediterranean diet (MedDiet) on CVD and its risk factors, including T2DM, has been a constant topic of interest. Notwithstanding, despite the large body of evidence, scientists are concerned about the challenges and difficulties of the application of MedDiet. This review aims to explore the motivations and challenges for using MedDiet in patients with CVD and T2DM. Design: An electronic search was conducted for articles about MedDiet published in PubMed, ScienceDirect, Scopus, and Web of Science up to December 2021, particularly on CVD and T2DM patients. From a total of 1536 studies, the final eligible set of 108 studies was selected. Study selection involved three iterations of filtering. Results: Motivation to apply MedDiet was driven by the importance of studying the entire food pattern rather than just one nutrient, the health benefits, and the distinct characteristics of MedDiet. Challenges of the application of MedDiet include lacking universal definition and scoring of MedDiet. Influences of nutritional transition that promote shifting of traditional diets to Westernized diets further complicate the adherence of MedDiet. The challenges also cover the research aspects, including ambiguous and inconsistent findings, the inexistence of positive results, limited evidence, and generalization in previous studies. The review revealed that most of the studies recommended that future studies are needed in terms of health benefits, describing the potential benefits of MedDiet, identifying the barriers, and mainly discussing the effect of MedDiet in different populations. Conclusions: In general, there is consistent and strong evidence that MedDiet is associated inversely with CVD risk factors and directly with glycemic control. MedDiet is the subject of active and diverse research despite the existing challenges. This review informs the health benefits conferred by this centuries-old dietary pattern and highlights MedDiet could possibly be revolutionary, practical, and non-invasive approach for the prevention and treatment CVD and T2DM.

## 1. Introduction

Nutritional transition accompanying globalization and urbanization has been pivotal in understanding the increased risks of overnutrition and its related conditions including overweight, obesity, and type 2 diabetes mellitus (T2DM) [1]. Lifestyle interventions such as promoting a healthy diet and increasing physical activity are important modifiable factors and are considered as means of preventing and treating non-communicable diseases (NCDs) [2,3]. Notwithstanding the considerable variation in the description of Mediterranean Diet (MedDiet) model used in difference studies, the Mediterranean diet is generally defined as a dietary pattern that emphasizes high intake of extra virgin olive oil, vegetables including leafy green vegetables, fruits, cereals (unrefined grains), nuts, pulses, and legumes, with moderate intakes of fish and other meat, dairy products and red wine, and low intakes of eggs and sweets [4]. Health benefits of MedDiet, with high consumption of extra virgin olive oil and nuts, have been repeatedly reproduced including its protective effects against the prevalence and progression of metabolic syndrome (MetS) [5], lower insulin resistance, arterial stiffness, blood pressure, serum lipids, and oxidative stress as well as lower sub-clinical atherosclerosis and cardiovascular disease (CVD) risk [6]. In many studies, MedDiet is negatively correlated with all-cause mortality [7]. The diet prevents and controls NCDs throughout patients’ lifespan [8]. Other positive findings include improvement in cognitive decline, Alzheimer’s disease, vascular dementia management [9], asthma, depression, colorectal cancer, and breast cancer [10]. 

Globally, CVD is highly prevalent. Although mortality rates of CVD are reducing in many countries, it remains as the world’s “number one killer” and the leading cause of disability in 2020 [11]. The role of MedDiet in CVD and its risk factors have gained much attention in recent decades, with numerous clinical trials and epidemiological studies highlighting the beneficial effects of the diet, attributed partly by its high composition of polyphenols and monounsaturated fatty acids or polyunsaturated fatty acids [12]. The American Diabetes Association recommends MedDiet for its beneficial effects on cardiovascular risk factors and glycemic control [13]. 

The Mediterranean diet pattern is associated with the prevention of non-communicable disease explained by inadequate nutrition and often advocated as a model of healthy eating. Studies consistently report that MedDiet can preserve good physical wellness and improve quality of life. Evidence regarding the protective effect of MedDiet on the risk factors of CVD is put forward, wherein the diet significantly reduces CVD mortality and morbidity [14]. The diet is deduced as an effective alternative strategy for glycemic control compared with a lower-fat, higher-carbohydrate dietary pattern for patients with T2DM or those with cardiovascular risk factors [15]. 

Although the benefits of MedDiet are widely established, concern remains as to whether the diet, which was originated and practiced by people living in Mediterranean basin, works well for people living beyond Mediterranean countries. Therefore, the aim of this review is to provide updates on the motivations and challenges of the application of Mediterranean diet in CVD and T2DM. 

## 2. Methodology

A systematic review of research articles published in the last 8 years (2014–2021) was conducted. We restricted our search to articles focusing on the motivations and challenges for MedDiet and its health-related benefits, including those with CVD and T2DM. 

### 2.1. Information Sources 

The search for target articles was conducted using four databases: (1) PubMed, (2) ScienceDirect, (3) Scopus, and (4) Web of Science. PubMed is a free resource that facilitates the search and retrieval of biomedical and life sciences literature with the objective of enhancing global and personal health. ScienceDirect is a huge database of medical and scientific publications published by the British publisher Elsevier. Scopus is the abstract and citation database of Elsevier, which launched in 2004. Nearly 11,678 publishers with 36,377 titles are included in Scopus. Web of Science provides subscription-based access with comprehensive citation data from multiple databases for many disciplines in the academic sector. The selection was made to present a relevant, broader view and with a range of disciplines in line with the efforts provided by the researchers.

### 2.2. Study Selection 

Three iterations of filtering were used in the study selection. In the initial version, redundant articles were eliminated. In the second iteration, irrelevant articles were deleted by examining titles and abstracts. After a thorough full-text reading of the screened articles from the first step, the third iteration filtered the articles. 

### 2.3. Search 

The search was carried out for publications from January 2014 to December 2021 in the databases of PubMed, ScienceDirect, Scopus, and Web of Science via the search box. A variety of keywords were used, including “Mediterranean diet”, “type 2 diabetes mellitus”, and “cardiovascular disease” in various combinations with the “OR” word. The exact query is shown in Figure 1.

### 2.4. Eligibility Criteria 

Articles were included on the basis of the criteria listed in Figure 1. We set an initial target for MedDiet consumption and the associated health benefits, which include CVD and T2DM. Articles that did not match the eligibility requirements were excluded from screening and filtering iterations after duplicates were removed. Exclusion reasons are (1) the article is written in a language other than English and (2) the target is other health benefits. 

The inclusion criteria are the articles in an English journal and those focusing on the MedDiet and its relationship with T2DM and CVD.

### 2.5. Data Collection 

A complete list of all selected articles along with their respective primary categories was collected from several sources into a single EXCEL file to facilitate subsequent steps. Two authors conducted extensive readings of the full text, resulting in a compilation of comments and highlights of the surveyed works. The text body was saved with all the comments. This step was followed by a summary, tabulation, and description of the main findings. The collected information, including the review sources, description tables, their respective source databases, summary, and the full list of articles, were saved as EXCEL and WORD files. 

## 3. Results 

From 2014 to 2021, the initial query researchers reviewed 1536 articles: 101 from PubMed, 741 from ScienceDirect, 421 from Scopus, and 273 from Web of Science. Sixty-seven duplicate articles were found across the four library databases. After screening the titles and abstracts, 890 articles were further excluded, resulting in 579 papers. Finally, 108 articles were included in the set after excluding 471 articles in the full-text reading.

## 4. Discussion 

This section aimed to explore and discuss the main elements of the MedDiet and its health benefits for CVD and T2DM, including motivations, challenges, and recommendations based on the current findings. The aim is to emphasize the current research trends in this scope. 

### 4.1. Motivations 

Mediterranean dietary pattern possesses a millenary tradition of use with absence of evidence of harm [16]. This section reiterates the health benefits of MedDiet and informs and persuades the adoption of MedDiet for the pursuit of healthcare sustainability. In general, there are three aspects that promote or motivate the application of MedDiet, including food pattern, which discusses the importance of studying the entire food pattern rather than just single nutrient, the established health benefits of MedDiet, specifically on CVD, T2DM, hypertension, and body weight management, and last but not least, the characteristics of MedDiet, including its adoptability, palatability, and safety that promote application of MedDiet. 

The rationale for recommending the use of MedDiet can be considered in the aspects of a shift in nutritional epidemiological research, practicality and feasibility, as well as sustainability of MedDiet. Traditionally, the vast majority of nutritional epidemiological studies investigating the associations of diet, health, and diseases have focused on single nutrients or single foods, with the evidence being reflected in current dietary recommendations. This decompositional or reductionist way, which highlights certain dietary constituents instead of dietary patterns, is aligned with an early study that helped to define the field of nutrition science through landmark discoveries linked to nutrient deficiencies [17] but presents several limitations and challenges. It is well-established that people consume meals consisting of a diversity of foods with complex combinations of nutrients that are likely to be interactive or synergistic [18], instead of isolated foods or nutrients. Studies on nutrition epidemiology has been extensive and dynamic. In recent years, studies shift from focusing on single nutrients or foods to evaluating the overall diet pattern [6,15,19,20,21,22,23,24]. Epidemiological studies that have focused on the role of single foods and/or individual nutrients have produced inconclusive findings regarding their influence on overall metabolic health [25]. Instead, evidence establishes that metabolic health is less influenced by single nutrients and more by intake of specific foods and overall dietary patterns [26]. In patients with diabetes mellitus, dietary patterns have a strong impact on specific cardiovascular risk factors [27]. The dynamic and extensive research evidence is translated into the move of single-nutrient recommendations toward beneficial dietary patterns as an approach for informing public health recommendations [28] as seen in the latest Dietary Guidelines for Americans 2020–2025. To date, there is also proposal to consider dietary patterns in developing future iterations of the Australian Dietary Guidelines [29].

Despite national dietary guidelines being available as early as 40 years ago, population dietary change has been slow, with many still falling short of current food-based dietary recommendations [30], and poor adherence to dietary guidelines is evident. This is partially attributed to the difficulties of translating present dietary recommendations into food-based public health advice [31] and lacking certain universally consistent recommendations across countries [32]. Compliance is further complicated with the expectation of certain level of health literacy such as calculation of calories and label reading among the socioeconomically deprived communities. Dietary guidelines that focus on dietary patterns rather than individual nutrient recommendations could help avoid confusion and avoid inadvertent increases in one nutrient of concern at the expense of another [33]. Growing evidence from intervention studies on MedDiet demonstrated generally a good compliance with a Mediterranean diet among the patients (need the complicate rate). The flexibility of MedDiet with the emphasis on encouraging positive behaviour change and improving dietary quality through displacement of discretionary items with healthy nutrient-dense foods instead of prohibiting of certain goods allow better compliance on MedDiet [34]. 

The notion of the MedDiet has undergone progressive evolution over half a decade to what we now observe as a dietary pattern with nutrition, food, cultures, people, environment, and sustainability all interacting into a new model that is sustainable. The flexibility of MedDiet that can be modified and accommodated to different cultural differences, food systems, and seasonal variations makes it transferable not only in Mediterranean basins, but in non-Mediterranean populations including multiethnic populations in Australia [34] and other countries. Global food demand is increasing rapidly with the presence of multiple drivers, including population growth, dietary shifts, and economic development. To satisfy the increasing global food demand within a resource-constrained planet is a daunting global challenge [35]. Meeting the rising global food demand to support people requires expanding agricultural production and is a major source of poor health and environmental degradation and represents the largest driver of biodiversity loss. It is important to acknowledge that as populations become more affluent and urbanized, this will further ignite the food demand, particularly for animal food sources, and increase environmental impacts. Hence, there is a need for a transition to more environmentally sustainable and healthier dietary patterns. Characterized by a plant-based diet, with little to moderate amounts of animal food sources, the Mediterranean Diet plays a pivotal role in supporting a healthy diet transition towards more sustainable agriculture and food systems. Adherence to the Mediterranean Diet has shown reductions in 19–43% of water footprints, 72% of greenhouse gas emissions, 58% of land use, 52% of energy consumption, and 33% of water consumption [36].

#### 4.1.1. Health Benefits

The Mediterranean Diet has various health benefits, particularly its responsibility in the prevention of diseases and improvement of health outcome such as CVD, T2DM, and their risk factors. 

##### Cardiovascular Diseases 

Most of the studies on primary prevention of CVD reported a statistically significant inverse association between MedDiet and the incidence of the disease [14]. The results were consistent in many other studies, supporting the evidence that MedDiet is protective against the risk of CVD [2,6,9,10,15,23,37,38,39,40,41,42,43,44,45,46,47,48,49,50]. MedDiet is also considered a healthy dietary pattern associated with decreased risk factors of CVD [51] and other health benefits [41]. In a recent critical review paper, MedDiet is recognized as a dietary pattern that has undergone the most comprehensive, repeated, and international assessment of its cardiovascular effects as compared to other dietary patterns [52]. It was identified as the most significant dietary pattern to provide protection against cardiovascular heart disease [13,53] as well as its progression [7]. Mediterranean countries such as Greece and Italy have a significant lower prevalence of cardiovascular heart diseases compared with other countries in the north of Europe or the US [45]. The potential advantages of MedDiet are related to the synergic effect and mechanisms of specific nutrients of MedDiet that have a direct impact on risk parameters of CVD including the oxidative stress, apolipoproteins, plasma lipid/lipoprotein profiles, vascular function, body weight/composition, inflammatory markers, glycemic response, and MetS components [46,54]. 

##### Type 2 Diabetes Mellitus 

Adherence to the MedDiet pattern results in a protection against cardiodiabesity [14]. In the last six years, the traditional MedDiet has gained popularity as a healthy dietary pattern linked to a lower risk of T2DM [3]. Consistently, greater adherence to MedDiet is linked to a statistically significant decrease in T2DM risk in many studies [6,15,40,55,56,57].

Prospective epidemiologic studies suggest that dietary patterns characterized by lowered intake of starchy foods, sugar-sweetened drinks, and red and processed meat and a high intake of whole grains, vegetables, fruits, and fish delay the onset of T2DM. Primary prevention of T2DM is facilitated by moderate and high adherence to MedDiet, which is mediated by enhanced total antioxidant capacity and reduced inflammation [41]. MedDiet pattern is high in monounsaturated fatty acids, which improve glycemic control in T2DM, implying that a high adherence promotes insulin sensitivity [6]. Similarly, the extensive use of olive oil in MedDiet can offset the adverse effects of inflammation, oxidative stress, and acute hyperglycemia on endothelial function and recover the protective action of GLP-1 on insulin secretion, endothelial function, and inflammation in T2DM [12]. This was imperative for the clinical management of T2DM due to the development of cardiovascular complications with a reduced protective effect of GLP-1 and acute hyperglycemia [12]. 

In particular, higher consumption of traditional MedDiet is correlated with decreased HbA1c levels and glucose levels two hours after eating [13,22,58,59]. The findings from several cohort studies reveal a significant inverse correlation between adherence to traditional MedDiet and plasma glucose as well as insulin concentrations among nondiabetic individuals. A systematic review and meta-analysis among 3073 individuals from 20 randomized control trials (RCTs) reveal that MedDiet, as well as a low-carbohydrate diet, reduces CVD risk effectively in T2DM cases [45]. MedDiet is also regarded as protective over the entire spectrum of nondiabetic and prediabetes participants [41]. 

The main physio-pathological process of T2DM is the state of sustained hyperglycaemia attributed to pancreatic β-cells impaired insulin secretion or/and cell insulin resistance [60]. The major role played by MedDiet is related to its glucometabolic benefits, including improvements in insulin resistance, insulin clearance, and β-cell function [61], which are possibly influenced by several factors including aging or genetic abnormalities [62]. In this regard, glucolipotoxicity [63], reactive oxygen stress [64], insulin resistance [62], activation of inflammatory pathways [65], amongst others, contribute to β-cell impaired insulin secretion. Improved glycaemic control is fundamental in T2DM complications prevention in both acute and chronic conditions. 

##### Risk Factors of Cardiovascular Disease and Type 2 Diabetes Mellitus

CVD and T2DM shared similar risk factors including excess body weight [66], excess fat accumulation [67], high blood pressure, [68] hyperglycemia [69], and dyslipidemia [70] or recently linked with the presence of MetS [71]. There is much existing literature available explaining the protective role of MedDiet against the risk factors in people without the diseases. This showed that MedDiet is able to prevent the preclinical condition of both CVD and T2DM before the onset of the diseases. Many studies reveal a significant benefit of adopting MedDiet in weight loss or maintenance [19,20,55] as a strategy for management of CVD and T2DM; however, some findings remain controversial. A meta-analysis of short-term RCTs of MedDiet for weight loss reports no effect on bodyweight when the diet is not calorie-restricted [72]. Nevertheless, the benefit of MedDiet on other aspects of weight management should not be neglected as evidence supports an inverse association between MedDiet adherence and the likelihood of obesity [15]. Gomez-Huelgas et al. (2015) reported that despite its relatively high-fat content, MedDiet prevents weight gain. Various characteristics of MedDiet, including high vegetable fiber and water content and low energy density and glycemic load, can help maintain weight [73], and this was also evident in the Seguimiento Universidad de Navarra (SUN) cohort among a Spanish population [74]. A Mediterranean diet helps to control patients’ blood pressure and further reduce long-term CVD complications for better management [6,45]. Moreover, MedDiet can reduce the odds of developing hypertension by 13% among non-diseased populations as well, which was evident in a systematic review and meta-analysis of RCTs. Therefore, MedDiet was suggested as an effective dietary strategy to control blood pressure and lower the risk of CVD [75]. An animal study reported that nutrients such as anthocyanins, flavanones, hydroxycinnamic acid, and ascorbic acid that are high in MedDiet are able to enhance anti-inflammatory activity and immune-modulation activity in hyperglycemic condition, which further prevent development of T2DM [76]. Adherence to a MedDiet was associated with better dyslipidemia and low-grade inflammation profiles in familial hypercholesterolemia [77], which may further reduce the risk of both CVD and T2DM.

##### Mortality of Cardiovascular Disease and Type 2 Diabetes Mellitus

Higher adherence to MedDiet is not only linked to the prevention and treatment of CVD, but also linked with a statistically significant decrease in all-cause mortality, CVD mortality, and cancer mortality as well as improvement in overall survival [78]. The Seguimiento Universidad de Navarra (SUN) cohort among a Spanish population revealed that high adherence to MedDiet is associated with a reduced incidence of all-cause mortality, fatal CVD, and non-fatal major CVD [75]. A meta-analysis justified that the reduced risk of all-cause mortality and CVD endpoints were associated with moderate alcohol intake as recommended in MedDiet [80]. The inverse relationship between the MedDiet and mortality is believed to be linked with the ageing biomarker—the telomere that is highly susceptible to inflammation-related and oxidation mechanisms. MedDiet, which traditionally rich in antioxidants or with marked anti-inflammatory properties, might reduce the telomere shortening rate, delay the ageing [79], and enhance the survival rates even with the presence of CVD or T2DM.

#### 4.1.2. Mediterranean Diet Characteristics 

MedDiet is easy to adopt, palatable, safe, and sustainable. Its flexibility [80] allows it to be easily adopted by different populations with various cultures [20,81,82]. It has beneficial effects on the primary and secondary prevention of several noncommunicable diseases (NCDs) [56]. Owing to these characteristics, MedDiet is recognized as the most healthy, environmentally sustainable, and economically affordable diet pattern, especially in Mediterranean countries, as they have higher accessibility of MedDiet components [14]. Attributed to its high sustainability and palatability, MedDiet is very useful as a preventive strategy for the optimal collaborative management of patients in primary care medicine [14]. It is mostly plant-based with high healthy fat content [2], which makes it palatable and easier to follow as compared with the low-fat diet [57]. The benefits of this diet are generally because of its whole components and not due to calorie restriction, weight loss, or increased physical activity [3]. Besides CVD and T2DM benefit, MedDiet also shows an inverse dose–response relationship with other health conditions including depression, cognitive decline, nephrolithiasis, and even fertility [74].

### 4.2. Challenges

Previous research has identified a number of challenges of the application of MedDiet. It is generally acceptable that there is absence of universal definition, adherence, and scoring of MedDiet, which greatly limits the ability to compare across studies and to translate the scientific research into practical recommendations for the general population [52]. Lacking a universal definition for MedDiet is attributable to the differences with regards to religious, political, economic, and social aspects within and between the more than 20 countries in the Mediterranean basin [22]. Several a priori-defined indexes with different scoring or coding systems in measuring adherence are available. The heterogeneity of the adherence scores raises the potential for disparity in statistical analyses and confusion as to which specific score should be chosen from. Variability across MedDiet scores and indices entails a potential for misclassification [83]. 

MedDiet has risen to fame because of its associated health benefits and continues to raise interest especially in light of the growing challenge of the prevalence of NCDs globally. Notwithstanding these well-documented health benefits and the low environmental impact of the MedDiet, recent surveys show a decline in its adherence in Northern, Southern, and Eastern Mediterranean countries, because of multifactorial influences—lifestyle changes, globalization of food markets, and sociocultural factors. This is especially true among the younger generations [84,85,86], attributed to nutritional transition experienced by the younger generations as characterized by shifting of traditional MedDiet to a “Western diet” rich in saturated fat, refined grains, and processed foods [87]. As nutritional transition is a universal recognized vector for increased prevalence of overweight, obesity, and diet-related non-communicable diseases, deviation from healthy dietary patterns has become new challenge for public health systems in Mediterranean countries. 

On the other hand, although the health benefits of MedDiet are well proven, researchers claimed that adopting the diet in non-Mediterranean regions continues to face uncertainty [45]. Adherence to MedDiet is a frequently raised concern, as it is regarded as difficult in a non-Mediterranean region [45]. One reason for such a claim is the existence of many factors, such as the composition of foods available in Mediterranean and non-Mediterranean countries [46], and the availability of food, which can affect the adherence to MedDiet [46]. Extra-virgin olive oil is a key feature of MedDiet, but its intake is particularly scarce in non-Mediterranean populations [88]. In a sample of middle-aged adults in the United Kingdom, several perceived barriers when attempting to adapt principles of the MedDiet were identified, including purchasing, organizing, and preparing food due to time constraint. Esposito et al. (2015) reported a potential issue, that is, the ability to translate this dietary pattern from the Mediterranean region to an Australian population who are ostensibly consuming a Western dietary pattern [89]. Notwithstanding, the latest national level adherence of MedDiet among Australia was reported to be medium (65.6%) or high (13.9%) [56], with comparable findings in Ireland [90,91] and Japan. Studies showed Scandinavian countries and other non-European countries recorded higher adherence to a Mediterranean-like dietary pattern when compared with the past [86,87]. The growing evidence allow us to assume MedDiet is increasing its popularity in non-Mediterranean countries. 

Diet cost is generally considered a determinant of food choices. Evidence indicates that healthy diets such as MedDiet cost more [92,93,94], which is associated with higher consumption of food components traditionally marked as healthy, namely fruits, vegetables, legumes, nuts, fish, cereals, and olive oil as compared to processed meats, potatoes, or sweets at lower dietary cost [94]. While the higher diet cost of MedDiet may influence food choices of an individual and limits its adherence, especially among the economically deprived population [95], cost-effectiveness analysis of MedDiet was consistently associated with favorable health outcomes [96,97,98] and better quality of life [94] and represents an exceptional return on investment [99]. Public health preventive strategies should consider keeping the Mediterranean diet pattern at a reasonable price and affordable for all people.

Reputed for its health effects, research on the effectiveness of MedDiet on health outcomes has spiralled, spanning from ameliorating risk factors of cardiovascular disease, [100] including glycemic and blood pressure control, to improved pregnancy outcomes [101,102], cognitive impairment [103], all-cause mortality [103], and others. On the other hand, a certain degree of controversy remains in the respect of some issues. Using blood pressure as an example, notwithstanding the growing evidence that MedDiet may improve endothelial function [104] and offer a considerable benefit in reducing blood pressure in hypertensive or healthy people, the scarcity of studies and the heterogeneity of the studies had restricted the ability to declare how strong the effect of MedDiet is [105]. In addition, effectiveness of MedDiet on blood pressures was less conclusive; inconsistency in findings were reported in observational or intervention studies [106], which may be attributed to the sample of the population or the short duration of intervention, respectively. It is imperative to emphasize that important methodological differences and limitations in the studies make comparison of results challenging. Against the large body of epidemiological observational studies, there is less evidence from well-conducted and adequately powered randomised controlled trials on the effectiveness of MedDiet. To the best of the authors’ knowledge, there were at least 10 meta or systematic analyses performed so far on the effect of MedDiet on blood pressure [107,108,109]. Concern exists in such meta-analysis or systematic analysis as the total trials could be small, the quality of included studies was suboptimal with high risk of bias, there was no universal similarity in the MedDiet trials, or there was a small magnitude of effect and high heterogeneity of participants, interventions, and comparators. Until now, no controlled clinical studies specifically evaluate the role of MedDiet in decreasing mortality in T2DM and cardiovascular events [22]. Only one large clinical trial is available, but it does not find any association between MedDiet and MetS [13]. Koloverou et al. (2014) reported the only RCT that assessed the role of MedDiet in the development of T2DM. Furthermore, only a handful of controlled trials designed to evaluate the cardiovascular and metabolic outcomes of MedDiet, specifically in T2DM patients, were available [15]. Large dietary intervention studies with long-term follow-up are required in the future to provide conclusive findings that a MedDiet pattern can reduce cardiovascular risk factors [49]. 

## 5. Conclusions

In conclusion, this review provides a comprehensive resume of the published studies in relation to the MedDiet CVD, T2DM, and their risk factor prevention, treatment, and outcome managements. This is the first review to add and collate significant new information from across a wide range of different studies by demonstrating the motivations and challenges in the application of MedDiet. Compared with previously available analyses, this review reinforces and more precisely identifies MedDiet as the subject of active and diverse research despite the existing challenges due to its various beneficial characteristics in managing the risk factors to prevent and manage CVD and T2DM. Available evidence in general support the health-related outcomes associated with adherence of MedDiet. Notwithstanding, there is mediocre robustness on the MedDiet, from the research and policy perspective. Despite several challenges to the implementation of MedDiet in non-Mediterranean countries and promoting the high adherence of MedDiet in Mediterranean countries, such barriers herein should not overlay the promising health effects of the centuries-old dietary pattern. Further investigation using long-term interventional as well as well-designed observational studies are warranted to inform nutrition recommendations in a wider context.

## Figures and Tables

**Figure 1 nutrients-14-02777-f001:**
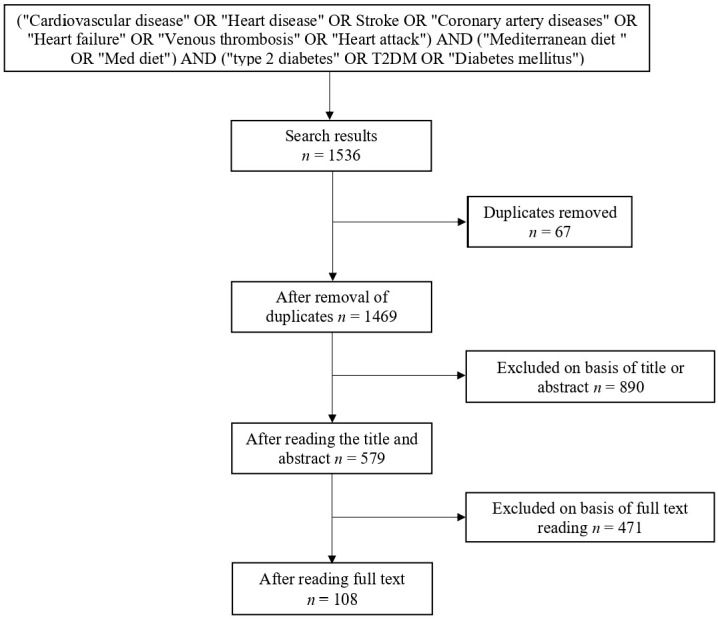
Flowchart of inclusion criteria, study selection, and search query.

## Data Availability

Data is contained within the article.

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
