# Peer review of "Application of Mediterranean Diet in Cardiovascular Diseases and Type 2 Diabetes Mellitus: Motivations and Challenges"

_nutrients, 2022, doi:10.3390/nu14132777_

Round 1

Reviewer 1 Report

Dear author,

Is a pleasure to review your manuscript, with an interesting topic as the relation of Mediterranean Diet in cardiovascular.

Introduction

T2DM and CVD are not defined previously on the text, it is on the abstract, but not in the text.

2.3 Search

Is defined the beginning of the search but not the end, should be stablish the period of time.

4.1 Motivations

“This is reductionist or …………………………. deficiencies” is a directed copy from the article (17). Should be adapted to avoid plagiarism.  

This reductionist, or decompositional, approach—emphasizing specific dietary constituents rather than dietary patterns—aligns with early research that helped to define the field of nutrition science through landmark discoveries related to nutrient deficiencies.

https://www.mdpi.com/2072-6643/10/5/571/htm?

References 18, 25 and 26 does not correspond with the information on the text. Should be replaced.

4.1.2.3 Risk factor…………..

“was also evident in the SUN” SUN is not defined before. Should be defined.

4.1.2.4 Mortality of cardiovascular………

“The Seguimiento Universidad de Navarra (SUN)” is defined after used the acronyms before. Should be changed.

Best regards,

Author Response

Reviewer 1

Introduction

T2DM and CVD are not defined previously on the text, it is on the abstract, but not in the text.

Thank you for the reminder, amendments were done based on suggestion.

Page 1, Introduction, line 3: type 2 diabetes mellitus (T2DM).

Page 2, Introduction, line 6: cardiovascular disease (CVD).

 2.3 Search

Is defined the beginning of the search but not the end, should be stablish the period of time.

The complete details for period of search are included.

Page 3, 2.3 Search, line 1: 2014 up to December 2021

4.1 Motivations 

“This is reductionist or …………………………. deficiencies” is a directed copy from the article (17). Should be adapted to avoid plagiarism.  

This reductionist, or decompositional, approach—emphasizing specific dietary constituents rather than dietary patterns—aligns with early research that helped to define the field of nutrition science through landmark discoveries related to nutrient deficiencies.

https://www.mdpi.com/2072-6643/10/5/571/htm?

Thank you for the comment and suggestion. We have changed it accordingly.

Page 4, 4.1 Motivations, paragraph 2, line 6.

References 18, 25 and 26 does not correspond with the information on the text. Should be replaced.

Thank you for highlighting this.

We have replaced the reference (18) accordingly. We removed ref 25 and 26 as the facts were repetitive with other sentences. On the other hand, there is a new reference add (no 28), which allow readers to understand how the research evidence has been translated into informing public health nutrition recommendation

4.1.2.3 Risk factor…………..

“was also evident in the SUN” SUN is not defined before. Should be defined.

Thank you for pointing this out, we apologize for the mistake.

Definition for SUN (Seguimiento Universidad de Navarra (SUN) cohort ) is made on the revised manuscript.

Page 7, line 5

4.1.2.4 Mortality of cardiovascular………

“The Seguimiento Universidad de Navarra (SUN)” is defined after used the acronyms before. Should be changed. 

Thank you for pointing this out, we apologize for the mistake.

Definition for SUN (Seguimiento Universidad de Navarra (SUN) cohort ) is made on the revised manuscript.

Page 7, line 5

Reviewer 2 Report

I read with great interest the paper “Application of Mediterranean diet in cardiovascular diseases and Type 2 diabetes mellitus: motivations and challenges" by Najwa Salim AlAufi.

The article is well written. The paper has a good design. The article is logically divided into sections and subsections.

Comments:

1.      Introduction: more should be said about Mediterranean Diet, as it is not mentioned in the text. In fact, the Mediterranean diet model is extremely variable among countries and regions due to culture, ethnicity, religious and agricultural habits. It commonly includes nutrition with mainly unrefined grains, vegetables and fresh fruits, olive oil, nuts, fish, white meats and legumes in moderation, while red meat, processed meat and sweets are limited, and red wine consumption is not excessive (doi: 10.3390/antiox10020270).

2.      4.1.2.2: The main physio-pathological process of type 2 diabetes is the state of sustained hyperglycaemia caused by pancreatic β-cells impaired insulin secretion and/or cell insulin resistance. The main role played by Mediterranean diet is the amelioration of both glycaemic control and increased insulin sensitivity. This is performed through an improvement in proper insulin secretion by β-cells, which may be influenced by several factors in addition to genetic abnormalities or aging. In this regard, lipotoxicity, glucotoxicity, reactive oxygen stress, activation of inflammatory pathways, IR leading to β-cell stress, and/or the decrease in incretin effect (GPL1 and GIP) on β-cells, amongst others, contribute to β-cell impaired insulin secretion. Improved glycaemic control is fundamental in type 2 diabetes complications prevention in both acute and chronic conditions (doi: 10.3390/nu12082236).

Author Response

Reviewer 2

Introduction: more should be said about Mediterranean Diet, as it is not mentioned in the text. In fact, the Mediterranean diet model is extremely variable among countries and regions due to culture, ethnicity, religious and agricultural habits. It commonly includes nutrition with mainly unrefined grains, vegetables and fresh fruits, olive oil, nuts, fish, white meats and legumes in moderation, while red meat, processed meat and sweets are limited, and red wine consumption is not excessive (doi: 10.3390/antiox10020270).

Thank you for the comment.  We had included a description on Mediterranean Diet in the introduction, which read as:

Notwithstanding the considerable variation in the description of Mediterranean Diet model used in difference studies, Mediterranean Diet is generally defined as a dietary pattern that emphasize

high intake of extra virgin olive oil, vegetables including leafy green vegetables, fruits, cereals (unrefined grains), nuts, pulses and legumes, moderate intakes of fish and other meat, dairy products and red wine, and low intakes of eggs and sweets

4.1.2.2: The main physio-pathological process of type 2 diabetes is the state of sustained hyperglycaemia caused by pancreatic β-cells impaired insulin secretion and/or cell insulin resistance. The main role played by Mediterranean diet is the amelioration of both glycaemic control and increased insulin sensitivity. This is performed through an improvement in proper insulin secretion by β-cells, which may be influenced by several factors in addition to genetic abnormalities or aging. In this regard, lipotoxicity, glucotoxicity, reactive oxygen stress, activation of inflammatory pathways, IR leading to β-cell stress, and/or the decrease in incretin effect (GPL1 and GIP) on β-cells, amongst others, contribute to β-cell impaired insulin secretion. Improved glycaemic control is fundamental in type 2 diabetes complications prevention in both acute and chronic conditions (doi: 10.3390/nu12082236).

Our highest appreciation to you to suggest a paragraph to summarize the pathophysiological process of T2DM and how MedDiet can augment the glycemic control.

We had included the suggestion into the manuscript, with slight modification on the sentences to minimise risk of plagiarism

Page 6, line 30

Round 2

Reviewer 1 Report

Dear author, 

Changes have been applied correctly.

Best regards,